# Development of three-colour FRET cascade for force sensing of the putative RIAM-vinculin interaction in fibroblasts
Conor A. Treacy [1,2], Tommy L. Pallett [2], Tam T. T. Bui [3], Simon P. Poland[2], Mark A. Pfuhl [1], Maddy Parsons [1] & Simon M. Ameer-Beg [2] ✉

Förster resonance energy transfer (FRET) enables the measurement of molecular interactions and conformational dynamics in biological systems. FRET-cascade, a multistep energy transfer system involving three fluorophores, enables spatial and temporal mapping of molecular interactions. Here, we leveraged FRET-cascade with time-correlated single photon counting fluorescence lifetime imaging microscopy (TCSPC-FLIM) to explore the putative interaction between Rap1-interacting Adaptor Molecule (RIAM) and vinculin in focal adhesions. We developed a novel three-fluorophore FRET-cascade system, validated using purified proteins, spectroscopic analysis, structural modelling, and negative-staining transmission electron microscopy (TEM). Putative RIAM-vinculin interactions were explored in vinculin knockout mouse embryonic fibroblasts and revealed that RIAM binds to the N-terminus of vinculin in focal adhesions. Vinculin tension-sensing constructs report average forces of 3.0 ± 0.3 pN per focal adhesion, consistent with its role in mechano-transduction. This work establishes FRET-cascade as a powerful approach for dissecting multicomponent protein interactions and force-sensing dynamics in live cells.

Förster resonance energy transfer (FRET) techniques applied in cell biology provide data on macromolecular function, structure, dynamics, and the local fluorophore environment[1–3]. FRET can be used as a quantitative proximity sensor between fluorescently labelled proteins with high sensitivity between 1 and 9 nm[3,4]. This enables the detection of direct intra- and intermolecular interactions by various modalities in fixed and living cells[5]. We employed fluorescence lifetime imaging microscopy (FLIM) using time-correlated single photon counting (TCSPC), which provides accurate and sensitive FRET measurements[6–8]. Recent advancements, including faster acquisition speeds, have enabled real-time imaging of dynamic biological processes[9–13]. This has facilitated the development of more sophisticated FRET systems, incorporating multiplexed and cascaded donor-acceptor pairs, which have been investigated at the single-molecule level using ratiometric measurements[14–16]. Three-fluorophore systems comprising two donor-acceptor pairs, wherein the first acceptor is the donor in the second FRET pair, have been demonstrated[15,17] (Fig. 1A). We are the first to describe this type of FRET model as Cascade-FRET, but similar variants have been used previously for ensemble measurements[17–19] and, more recently, in single-molecule applications[15,20,21]. The sequential action of cascading multiple FRET interactions has been previously used to investigate multiple conformational states of a single protein[5,22]. Cascade-FRET is not just limited to determining whether an interaction has occurred[14,15]. Recent studies have shown that attachment of three fluorescent dyes to a single protein[14,21,23,24] or nucleic acid[1,15,25] enables the determination of the three-dimensional conformation, orientation, and activation state during dynamic processes such as protein folding, ligand binding, or post-translational modification. Furthermore, cascade-FRET can uniquely elucidate the order in which multi-component complexes assemble, interact, evolve, and ultimately disassemble.

Focal adhesions (FA) are ideally suited to the application of cascade-FRET. FA are large macromolecular assemblies that anchor cells to the extracellular matrix (ECM) and are the primary site of mechanical force transduction[26,27]. FA consist of integrins, which are heterodimeric trans-membrane receptors that directly connect to the ECM via their extracellular ligand-binding domains, as well as numerous intracellular proteins organised into layers which regulate force transduction and interaction with the cytoskeleton[28,29]. Proteins such as talin and kindlin can bind directly to integrin cytoplasmic domains, in part mediated through Rap1-induced membrane-associated protein (RIAM), and mediate recruitment of adaptor and signalling proteins, including vinculin, FAK, and paxillin, with dynamic

[1]Randall Centre for Cell and Molecular Biophysics, King's College London, London, UK. [2]Comprehensive Cancer Centre, School of Cancer and Pharmaceutical Sciences, King's College London, London, UK. [3]Institute of Pharmaceutical Science, School of Cancer and Pharmaceutical Sciences, King's College London, London, UK. ✉e-mail: simon.ameer-beg@kcl.ac.uk

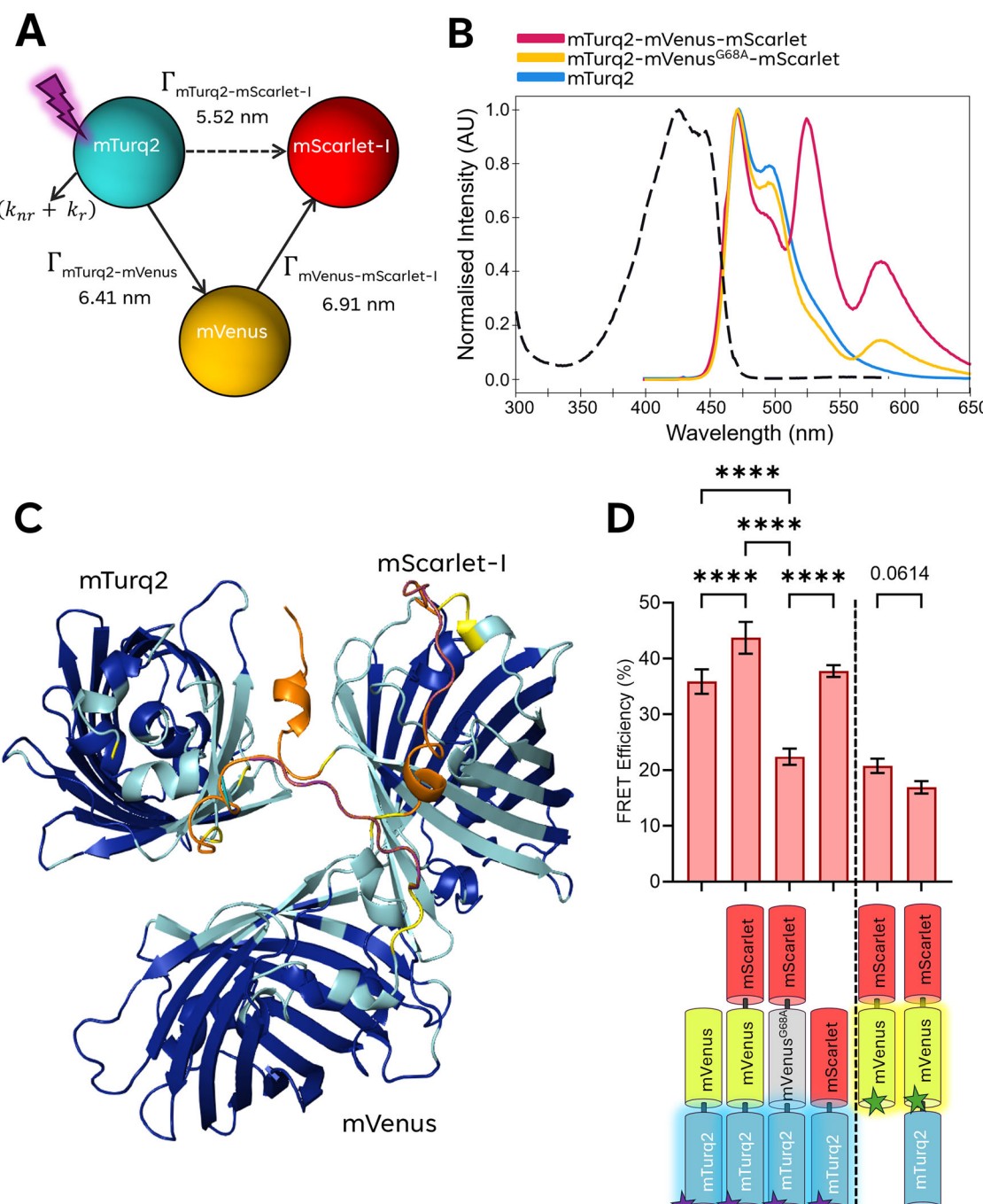

**Fig. 1 | Three-colour FRET-Cascade construct. A** A graphical representation of the fluorophores used in the three-colour FRET cascade module with specific energy transfers ($\Gamma$) labelled. The purple lightning bolt represents excitation at 435 nm. Solid arrows show the direction of energy cascade through mVenus, dashed arrow shows mTurq2 to mScarlet-I direct energy transfer. **B** Excitation spectra for mTurq2 and the emission spectra for mTurq2 (blue), mTurq2-mVenusG68A-mScarlet-I (yellow), and mTurq2-mVenus-mScarlet-I (red), where mTurq2 alone is excited at 435 nm (excitation spectrum indicated by black dashed line). **C** An AlphaFold3 model of the three-colour protein complex, regions with a high predicted local distance difference test (pLDDT) score >90 are dark blue, 70 > 90 light blue, 50 > 70, yellow and those with a very low pLDDT score ( < 50) are indicated in orange. **D** Average FRET efficiencies for the various FRET pairs measured using our custom TCSPC-FLIM system, n = 9 individual cells measured per condition across $N = 3$ technical repeats. Bars are the mean FRET efficiency ± SEM (error bars); comparisons of means are calculated by one-way ANOVA using Dunnett T3 correction for multiple comparisons: : $P \geq 0.123$ ns (ns), $P \leq 0.0332$ (*), $P \leq 0.0021$ (**), $P \leq 0.0002$ (***), $P \leq 0.0001$ (****), $p = 0.0614$ (mVenus-mScarlet vs. mTurq2-mVenus-mScarlet-I).

linkage to the cytoskeleton[30–35]. Previous studies have analysed tension across specific FA proteins to explore the relationship between mechanosensing, FA dynamics and cell migration[36]. The first example of this used a tension-sensing module (TSMod) containing a 40-amino acid long elastic domain integrated into vinculin between the Vh and Vt domains, creating a

vinculin-tension sensing (VincTS) construct[36,37]. Analysis of the VincTS construct in living cells using FLIM revealed that the average force in stationary FA was approximately 2.5 pN[36]. A putative binding interaction between the focal adhesion protein vinculin and the adaptor protein RIAM was identified in vitro between the N-terminus of RIAM (amino acids

https://doi.org/10.1038/s42004-025-01849-9                                                                    **Article**

1–127) and the N-terminus of vinculin (1–258), which was approximately five-fold weaker than the mutually exclusive vinculin-talin interaction[38]. However, the presence of this complex and its function in signalling or mechanosensing in FA remains unclear.

In the present study, we developed a three-colour fluorescent protein FRET-cascade to analyse tripartite protein interactions in live cells, consisting of mTurquiose2[39], mVenus[40], and mScarlet-I[41]. The photophysical properties of this construct were tested in vitro and behaved as predicted; this data was used to reproduce the model structure both with purified protein in vitro and in cells. Furthermore, introducing a glycine-to-alanine mutation in mVenus at amino acid 68 (Gly68Ala) provided a photochromically dead form of the fluorophore as an optimal control, which maintains stoichiometry and structural consistency. We applied this novel FRET-cascade to investigate the activation and binding of the mechanosensitive protein vinculin with its putative interactor RIAM (Rap-1a Interacting Adaptor Molecule). We characterise the interaction of full-length, fluorescently tagged vinculin and RIAM using transiently transfected vinculin knockout Mouse Embryonic Fibroblasts[36,42] (MEF$^{Vinc-/-}$) with TCSPC-FLIM. To explore the mechanical role of vinculin, the previously published vinculin tension sensor[37,43] (vincTS) was used to measure intracellular forces. We then applied the three-colour FRET-cascade model to describe the spatial relationship between vinculin and RIAM in FA. This work establishes a robust FRET-based approach for studying molecular interactions and intracellular forces. By successfully quantifying RIAM-vinculin binding and vinculin tension, the study provides new insights into focal adhesion mechanics and paves the way for broader applications in mechanobiology.

## Results

### Three-colour FRET cascade implementation

The generalised theory of stepwise and cascaded FRET efficiency was outlined by *Watrob* et al.[16]; we will describe only the pertinent elements (Fig. 1a). The measured donor fluorescence lifetime, $\tau_{DA}$, in the presence of an additional FRET decay pathway from the excited state, can be generalised as

$$\frac{1}{\tau_{DA}} = \Gamma_{DA} + (k_r + k_{nr}) = \Gamma_{DA} + \frac{1}{\tau_D} \quad (1)$$

where $\Gamma_{DA}$ is the FRET transfer rate from donor to acceptor, $k_r$ and $k_{nr}$ are the radiative and non-radiative decay rates for the fluorophore, respectively, and $\tau_D$ the donor lifetime in the absence of FRET. This can be rearranged to give generalised FRET efficiency for donor-acceptor pairs

$$E_{D_i A_j} = \frac{\Gamma_{D_i A_j}}{\Gamma_{D_i A_j} + \tau_D^{-1}} \quad (2)$$

Where the indices $i$ and $j$ indicate the position of the donor and acceptor in the cascade. Such a system is readily scalable and may contain any number of donors and acceptors. For any such system, total FRET efficiency measured for a given donor, $D_i$ is dependent on a sum over $n$ acceptors:

$$E_{D_i} = \frac{\sum_{j=1}^{n} \Gamma_{D_i A_j}}{\sum_{j=1}^{n} \Gamma_{D_i A_j} + \tau_D^{-1}} \quad (3)$$

Calculating the individual transfer rates allows us to extract the separation of donor and acceptor pairs directly. Given that the transfer rate is determined as

$$\Gamma_{D_i A_j} = \tau_D^{-1} \left(\frac{R_0}{r_{D_i A_j}}\right)^6 \quad (4)$$

Where $R_0$ is the Förster radius and $r_{D_i A_j}$ the molecular separation between $D_i$ and $A_j$. It is possible to dissect the molecular separations not simply on a pairwise basis but, in the presence of multiple interactions, to determine the coordinates of each molecule relative to the donor based on the modified lifetime of $D_i$ alone, given suitable controls. Such a system is generalisable to any number of fluorophores but quickly becomes unwieldy due to error propagation. In this paper, we restrict the experimental verification to three fluorophores.

A three-colour model to demonstrate the FRET-cascade principle was designed, consisting of mTurquoise2 (mTurq2)[44], mVenus[40] and mScarlet-I[41] (Fig. 1a). For each purified protein, excitation and emission spectra are given in Supplementary Fig. S1. Each fluorescent protein is separated by a flexible 6-amino acid linker (GGSGGS). Given the flexibility of the six-amino acid GGSGGS linkers and the rotational freedom of the attached fluorescent proteins (FPs), we assumed an orientation factor ($\kappa^2$) of 2/3 for Förster radius ($R_0$) calculations, consistent with the dynamic isotropic limit. Fluorophores act as a donor and/or acceptor within an energy transport cascade. mTurq2 was chosen for its high quantum yield (0.93) and monoexponential fluorescence emission lifetime of ~4.1 ns (in vitro solution)[39]. mVenus is well documented as an excellent FRET acceptor for mTurq2[39,45], and mScarlet-I, due to its high absorption cross-section, was chosen as an acceptor to either mTurq2 or mVenus[39,45]. mScarlet-I has substantial spectral overlap with mVenus and mTurq2, with an advantageous Förster radius ($R_0$) and brightness compared to other red FPs[39]. Essential metrics for selecting fluorophores include the Förster radius and the quantum yield of the first acceptor, as these factors ultimately limit stepwise interactions. Optimising these parameters is particularly advantageous when applying the FRET cascade to living cells (Supplementary Fig. S2 and Table S1).

A full range of pairwise constructs was purified, and the single FP spectra were compared with the mTurq2-mVenus, and mTurq2-mScarlet-I constructs (Supplementary Fig. S3). The excitation spectra for mTurq2-mVenus are similar to those for mTurq2 alone because only the mTurq2 is excited at 435 nm. However, the emission spectra differ, particularly at ~530 nm, corresponding to the peak emission of mVenus. The mTurq2-mVenus construct shows no spectral features corresponding to mScarlet-I, as expected. Since mVenus is minimally excited at 435 nm, the emission peak at 530 nm must result from mVenus FRET sensitised emission, with the magnitude of acceptor emission proportional to the FRET efficiency between the two FPs. A similar pattern is observed in the mTurq2-mScarlet-I protein, with the relative heights of these peaks corresponding to the FRET efficiency of the interaction. The emission spectra of mTurq2 alone are compared with those of the three-colour protein mTurq2-mVenus-mScarlet-I (Fig. 1b). Upon excitation at 435 nm, the emission spectra of the three-colour protein display peaks corresponding to all three reference spectra. The prominent peak at 530 nm indicates FRET between mTurq2 and mVenus. An additional peak at approximately 590 nm, corresponding to mScarlet-I, indicates a secondary FRET transition, where energy is transferred from either mTurq2 or mVenus to mScarlet-I, albeit with lower coupling efficiency. The presence of mVenus increases the separation between mTurq2 and mScarlet-I, reducing coupling via FRET compared to the mTurq2-mScarlet-I construct in the absence of mVenus.

A structural control containing a glycine-to-alanine mutation in the fluorochrome of mVenus designated mTurquoise2-mVenus$^{G68A}$-mScarlet-I, was cloned and purified. This mutation prevents mVenus from forming a functional fluorochrome (Supplementary Methods), thereby serving as an absorption/emission null structural control for the FRET-cascade model, while still properly folding and maintaining the overall structure. Spectral and circular dichroism measurements for mutated mTurquoise2-mVenus$^{G68A}$-mScarlet-I demonstrate the ablated mVenus fluorescence whilst remaining folded and retaining mScarlet-I sensitised emission (Fig. 1b, Supplementary Figs. S4 and supplementary methods).

The three-colour FRET-cascade construct was modelled using AlphaFold3[46] (Fig. 1c) using the single full-length sequence of the three-colour peptide. AlphaFold3 consistently predicted a triangular 3D tertiary structure formed by the β-barrels of the three FPs, though their relative

**Table 1 | Three-colour FRET model Lifetimes and FRET efficiencies**

| Purified Protein | Average lifetime (ns) | SEM | Average $\chi^2$/d.f. | | |
|---|---|---|---|---|---|
| **mTurq2** | 4.178 | 0.01177 | 0.949 | | |
| **mVenus** | 3.020 | 0.03676 | 1.095 | | |
| **mTurq2**-mVenus | 2.668 | 0.04371 | 0.908 | | |
| **mVenus**-mScarlet | 2.393 | 0.02838 | 0.991 | | |
| **mTurq2**-mScarlet | 2.600 | 0.01738 | 1.089 | | |
| **mTurq2**-mVenus$^{G68A}$-mScarlet | 3.242 | 0.03438 | 1.159 | | |
| **mTurq2**-mVenus-mScarlet | 2.332 | 0.05397 | 1.058 | | |
| mTurq2-**mVenus**-mScarlet | 2.509 | 0.06745 | 1.075 | | |
| **FRET Pair** | **FRET (%)** | **SEM** | **$R_0$ (nm)** | **R (nm)** | **SEM** |
| **mTurq2**-mVenus | 35.89 | 0.564 | 5.83 | 6.41 | 5.23E-02 |
| **mTurq2**-mVenus-mScarlet | 43.72 | 0.892 | n.a. | n.a. | n.a. |
| **mTurq2**-mVenus$^{G68A}$-mScarlet | 22.41 | 0.4572 | 5.08 | 6.25 | 5.49E-02 |
| **mTurq2**-mScarlet | 37.77 | 0.2739 | 5.08 | 5.52 | 2.15E-02 |
| **mVenus**-mScarlet | 20.78 | 0.7336 | 5.53 | 6.91 | 1.03E-01 |
| mTurq2-**mVenus**-mScarlet | 16.94 | 0.6397 | 5.53 | 7.21 | 1.09E-01 |

Summary tables detailing the fluorescence lifetimes and $\chi^2$/degrees of freedom (goodness of fit metric) for the purified proteins used in the FRET cascade model. FRET efficiencies, Forster radii ($R_0$), separation distance (R) and their respective SEM are also presented. $N = 9$ measurements, cells per condition across three separate technical repeats.

orientations remain undefined. For further details and model validation, see Supplementary Figs. S5, S6, S7 and the supplementary methods. To test that the cascade approach is structurally donor-agnostic, we exchanged the mTurq2 fluorescent protein for mTFP1 in silico. After superposition on the middle barrel, both the mTurq2 and mTFP1 constructs show N → M and M → C separations within FRET-compatible ranges, and the first-barrel folds match by CEalign. Hence, the choice of mTurq2 in this demonstration is purely practical, albeit the findings are readily extrapolatable to mTFP1, which is a widely used biosensor donor. For FLIM, mTurq2 has a longer lifetime and therefore improved dynamic range for FRET; this is a measurement consideration rather than a design requirement.

The three-colour proteins, mTurq2-mVenus-mScarlet-I and mTurq2-mVenusG68A-mScarlet-I, exhibited reductions in fluorescence lifetime compared to their single FP controls (Fig. 1d, Supplementary Fig. S8). However, mTurq2-mVenus$^{G68A}$-mScarlet-I had a longer fluorescence lifetime, consistent with the lack of a functional mVenus acting as an intermediary in the FRET-cascade. Average lifetimes, FRET efficiencies and $R_0$ values are detailed in Table 1. FRET efficiency for the mTurq2-mVenus protein increased with the addition of mScarlet-I, indicating FRET between mTurq2 and mScarlet-I.

Energy transfer rates were calculated from fluorescence lifetime data, allowing modelling of a theoretical three-colour FRET exchange (Table 2). The calculated FRET transfer rates correspond to distances and average angles between the arms formed by mVenus with mScarlet and mTurq2 that are consistent with the AlphaFold3[47] model prediction of a triangular configuration (Fig. 1c). However, the absolute values of the distances are shorter in the predicted structure from the AlphaFold model, likely due to not implementing Amber relaxation to correct for steric hindrances[48] and because AlphaFold does not account for solvation. Both factors result in a model that is more compact than the native setting. These findings, while not conclusive, provide validation for the proposed model (Supplementary Fig. S9).

Negative Staining Transmission Electron Microscopy (TEM) was used to obtain low-resolution ( < 1 nm) structural images of the three-colour mTurq2-mVenus-mScarlet-I protein (Supplementary Fig. S10) to validate the FRET-derived intramolecular distances. The TEM images reveal a geometry that matched well with the FRET model, and the distance and angle values agreed within the standard error. This agreement is notable given the simplicity of the FRET model, though some discrepancies may arise from the protein's orientation on the TEM grid. A comparison of the

**Table 2 | Three-colour FRET model energy transfers**

| | Energy Transfer ($s^{-1}$) | SEM | Lifetime (ns) |
|---|---|---|---|
| $K_{mTuq2}$ | 2.39E + 08 | 2.81E-08 | 4.178 |
| $K_{mTurq2-mVenus}$ | 3.75E + 08 | 1.64E-07 | 2.668 |
| $\Gamma_{mTuq2-mVenus}$ | 1.35E + 08 | 5.91E-08 | 7.382 |
| $K_{mTuq2-mVenusG68A-mScarlet-I}$ | 3.08E + 08 | 1.06E-07 | 3.242 |
| $\Gamma_{mTuq2-mScarlet-I}$ | 6.91E + 07 | 1.40E-07 | 14.471 |
| $K_{mTuq2-mVenus-mScarlet-I}$ Calculated | 4.44E + 08 | 2.32E + 07 | 2.253 |
| $K_{mTuq2-mVenus-mScarlet-I}$ Measured | 4.29E + 08 | 1.47E + 07 | 2.331 |
| Difference (%) | | | 3.36% |

A summary table detailing the energy transfer rates, SEM, and the associated lifetime of each energy transfer in the construct. K = energy transfer rate for a specific fluorophore; $\Gamma$ = specific FRET energy transfer within a FRET pair. $N = 9$ measurements cells per condition across three separate technical repeats.

distances and angles between the calculated FRET, AlphaFold3 model and TEM measurements are detailed in supplementary Table S2.

## Exemplification of the FRET cascade system in a biologically relevant context

Energy transfer rates obtained from the model FRET cascade can be used to determine changes in distances between adjacent biological molecules in a dynamic complex. We employ this method to validate our hypothesis that RIAM binds to vinculin under tension in cells and to map changes in applied force to vinculin upon association with RIAM. Whilst the RIAM-vinculin interaction was previously observed in vitro[38], confirmation in vivo and a mechanistic description are absent from the literature. Our AlphaFold3 model of the interaction between amino acids 1–30 of RIAM and autoinhibited full-length vinculin (Fig. 2a, Supplementary Figs. S11 and S12) supports published in vitro analytical gel filtration data[38]. Our AlphaFold3 model provides further evidence that RIAM and vinculin interact in isolation and when vinculin is autoinhibited.

The conformational states of vinculin are proposed to be spatially compartmentalised, with the autoinhibited (closed) state predominantly

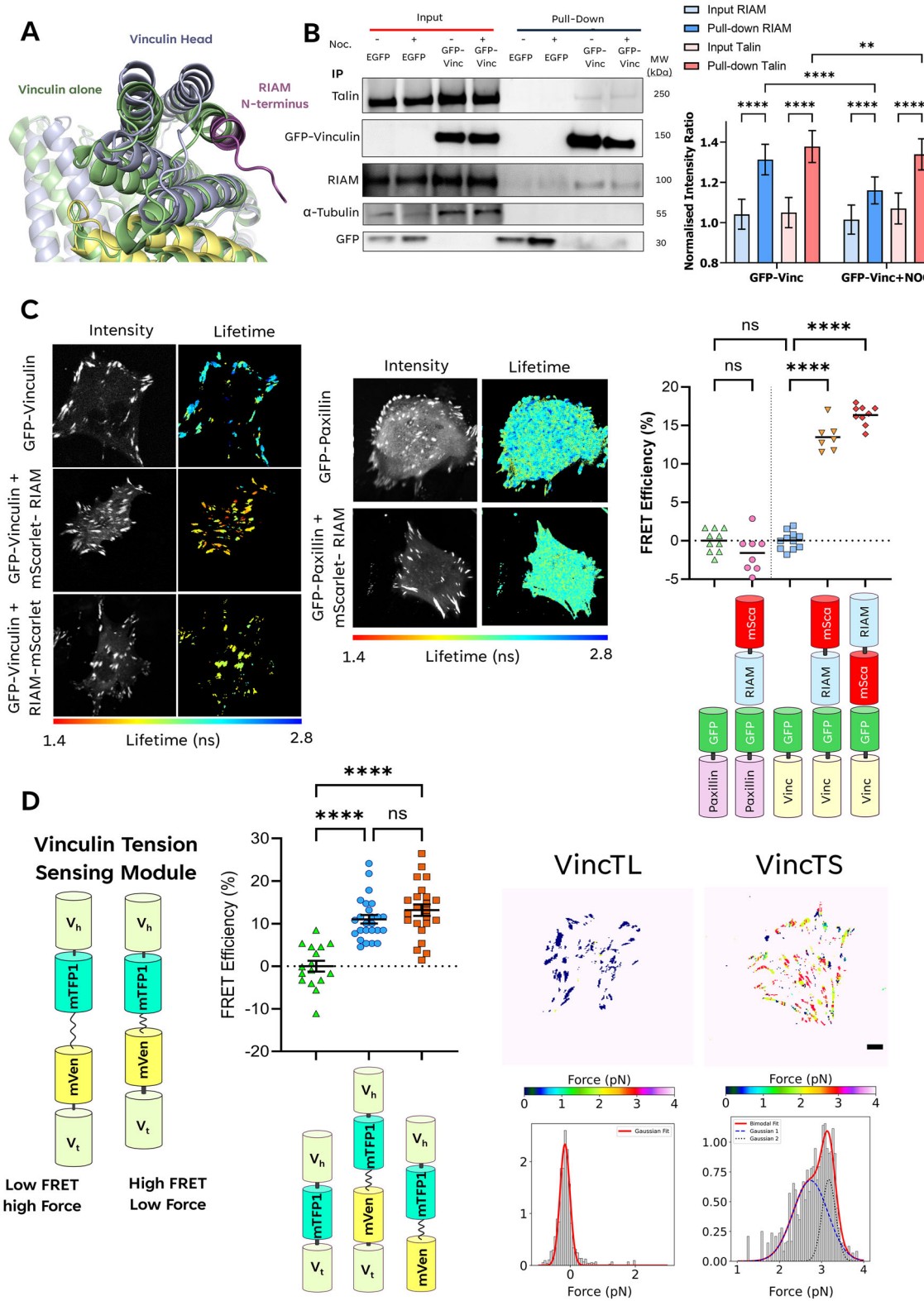

cytoplasmic and the active (open) state at FA, where it is under tension while bound to talin and actin[49–53]. Talin is a well-established binding partner of vinculin[52–56], RIAM and vinculin share a binding site on talin, implying that binding of both proteins is mutually exclusive. However, immunoprecipitated GFP-Vinculin expressed in MEF$^{Vinc-/-}$ formed a complex with both RIAM and talin (Fig. 2b). This demonstrates

that these proteins are sufficiently close to be isolated biochemically from cells, and that vinculin and RIAM may form a complex distinct from that with talin. Moreover, levels of RIAM—but not talin—were reduced in this complex following treatment of cells with the microtubule-disrupting agent nocodazole (Fig. 2b), which also stabilises FA[57]. This is consistent with previous reports of lower RIAM levels in mature FA[38], and suggests

**Fig. 2 | Establishment of the putative RIAM-vinculin interaction. A** An Alpha-Fold3 model of the RIAM-vinculin interaction. **B** A western blot of the co-immunoprecipitation of EGFP-vinculin with RIAM and Talin, probed with an antibody against EGFP. Noc. = Nocodazole treatment. To the right of the blot is a quantification of the pixel intensities of the protein bands for Talin and RIAM pulled down by GFP-Vinculin. The bar graph shows the mean intensity normalised to total protein, with error bars representing the standard error of the mean. Data taken from $N = 3$ blots; the blot displayed is a representative example. **C** Fluorescence lifetime data for EGFP-vinculin ± mScarlet-RIAM/RIAM-mScarlet and EGFP-Paxillin ± mScarlet-RIAM with average FRET efficiencies plot alongside. **D** Fluorescence

lifetime data for vincTS and controls. The right-hand panel shows typical FRET efficiency distributions and associated tension distributions for a vincTL or vincTS transfected MEF$^{Vinc-/-}$. Above are colour-scaled force maps; scale bar = 5 μm. Below are the histograms of the pixelwise forces, fitted to distributions as described in the main text. Red line = Gaussian fit; Blue line = 1st Bimodal fit; Black line = 2$^{nd}$ Bimodal fit. Data points in the graphs represent the average FRET efficiency per cell, and the bar represents the pooled population mean. Comparisons of population means are calculated using one-way ANOVA with Tukey's correction for multiple comparisons: P ≥ 0.123 ns (ns), P ≤ 0.0332 (*), *P ≤ 0.0021 (**), P ≤ 0.0002 (***)*, P ≤ 0.0001 (****).

a reduction of the RIAM-vinculin complex in favour of talin-vinculin binding.

To further investigate the RIAM-vinculin interaction, FRET efficiency distributions were obtained from histograms of the intensity-weighted lifetimes of cells expressing EGFP-vinculin with or without the co-expression of RIAM N- or C-terminally tagged with mScarlet-I (Fig. 2c). As expected, co-expression of mScarlet-tagged RIAM showed increased FRET efficiency and decreased EGFP lifetime (Supplementary Fig. S13). Notably, a higher FRET efficiency (16.4 ± 0.4%) was observed for N-terminally tagged RIAM compared to C-terminal (13.5 ± 0.7%), consistent with previous in vitro evidence[38] and supporting the AlphaFold3 predictions (Fig. 2a) that the interaction occurs between the N-termini of both proteins. Furthermore, FRET analysis revealed that this interaction also occurs outside of FA (Supplementary Fig. S14).

The dependence of the RIAM-vinculin complex on FA maturation state indicated a potential role for tension-dependent vinculin conformation in the control of RIAM binding. To address this, we leveraged a previously developed vinculin intramolecular tension sensor (vincTS) comprising a mTFP1 donor fluorophore and mVenus acceptor fluorophore linked by a 40-amino acid flagelliform linker (TSMod) inserted into full-length vinculin[36,37,58]. Vinculin adopts an open conformation once bound to both talin and F-actin and under tension[26,36,49,50,55,59]. The same tension-sensing module, lacking the C-terminal actin-binding domain (vincTL), was used as a high FRET, no-tension control. High variation in mean FRET efficiencies was seen in both FA-containing vincTS and those containing vincTL between different cells (Fig. 2d and Supplementary Fig. S15). Previous experiments using the same tension sensor demonstrated static FA with an average force of ≈ 2.5 pN[36]. As individual vinculin proteins bind within adhesions and diffuse in and out of the FA complex, we expect a full range of possible FRET values. The average force applied per pixel can be calculated[36,37]:

$$\langle F \rangle = (\langle R_{TS} \rangle - \langle R_{TL} \rangle)(0.01196\,N + 0.0001255) \quad (5)$$

Where $R_{TS}$ and $R_{TL}$ are the calculated separation distances for the vincTS and vincTL constructs, respectively, and $N$ is the number of amino acids in the linker. Both in vitro and in vivo studies[36,37,43,58] using the TSMod-based sensor, the 40-amino-acid linker used in the vincTS and vincTL constructs was found to be elastic, with an intracellular compliance of approximately 0.478 nm·pN$^{-1}$. From these FRET data, it is possible to calculate from Eq. (5) an average intracellular tensile force $\langle F \rangle = 3.0 \pm 0.3$ pN. A representative tension map for a cell expressing the VincTS construct, fitted with a bimodal Gaussian distribution, illustrates two populations with different levels of tension across vinculin in FAs. Low and high intramolecular tension across FL vinculin were $x1 = 2.7 \pm 0.4$ and $x2 = 3.2 \pm 0.2$ pN, respectively (Fig. 2d and Table 3).

To investigate RIAM-vinculin interactions as a function of force across vinculin, vincTS was expressed ± mScarlet-RIAM in MEF$^{Vinc-/-}$ cells for analysis using the developed FRET cascade (Fig. 3a). mTFP1-vinculin + mScarlet-RIAM exhibited an average FRET efficiency of 20.0 ± 3.0% (Fig. 3b), similar to that seen with GFP-vinculin (Fig. 2C). Co-expressing mScarlet-RIAM and vincTS significantly increased the average FRET efficiency from 12.3 ± 6.0% to 29.1 ± 8.8% (Fig. 3b). However, FRET efficiency of vincTL also increased from 17.0 ± 5.7% to

**Table 3 | Comparison of Bimodal fitting parameters for VincTS and VincTS + mScarlet-RIAM using the Cascade FRET energy transfer model**

| Lifetime and force bimodal fit parameters for VincTS only | | | Lifetime and force bimodal fit parameters for VincTS + mScar-RIAM | | |
|---|---|---|---|---|---|
| **Lifetime Bimodal Fit Parameters:** | | | **Lifetime Bimodal Fit Parameters:** | | |
| | **Lifetime (ns)** | **Std. dev.** | | **Lifetime (ns)** | **Std. dev.** |
| $x_1$ | 1.893 | 0.039 | $x_1$ | 2.154 | 0.182 |
| $\sigma_1$ | 0.249 | 0.026 | $\sigma_1$ | 0.437 | 0.121 |
| $A_1$ | 0.838 | 0.080 | $A_1$ | 0.584 | 0.993 |
| $x_2$ | 2.547 | 0.072 | $x_2$ | 3.146 | 0.270 |
| $\sigma_2$ | 0.305 | 0.051 | $\sigma_2$ | 0.406 | 0.181 |
| $A_2$ | 0.571 | 0.048 | $A_2$ | 0.362 | 0.118 |
| **Forces Bimodal Fit Parameters:** | | | **Forces Bimodal Fit Parameters:** | | |
| | **Force (pN)** | **Std. dev.** | | **Force (pN)** | **Std. dev.** |
| $x_1$ | 2.737 | 0.080 | $x_1$ | 3.067 | 0.0583 |
| $\sigma_1$ | 0.413 | 0.046 | $\sigma_1$ | 0.409 | 0.0394 |
| $A_1$ | 0.678 | 0.065 | $A_1$ | 0.784 | 0.0666 |
| $x_2$ | 3.172 | 0.019 | $x_2$ | 3.454 | 0.0219 |
| $\sigma_2$ | 0.161 | 0.030 | $\sigma_2$ | 0.114 | 0.0292 |
| $A_2$ | 0.686 | 0.140 | $A_2$ | 0.633 | 0.1345 |

A summary table detailing the lifetime. Standard deviation and fractional intensity for each peak in the bimodal modal for lifetime and force for VincTS and for VincTS + mScarlet-RIAM. Where $x$ = the mean; $\sigma$ = standard deviation, and A = amplitude of the bimodal Gaussians.

28.9 ± 3.3% with the addition of mScarlet-RIAM (Supplementary Fig. S16), indicating FRET between mTFP1 and mScarlet can occur in the presence and absence of vinculin-F-actin binding. Further analysis in cells treated with the ROCK inhibitor (H1152) to prevent actomyosin contractility showed no change in FRET efficiency of the RIAM-vinculin interaction (Supplementary Fig. S17). This would imply that RIAM binds to vinculin in a force-independent manner.

To determine if VincTS was under tension when mScarlet-RIAM was present, the energy transfer rate contribution between mTFP1 and mScarlet was calculated and substituted into Eq. (3) to calculate the FRET rate independent of mScarlet-RIAM. This allowed us to determine tension levels across VincTS in the presence of mScarlet-RIAM and compare them with those in its absence. The tensile force for a typical single cell expressing vincTS + mScarlet-RIAM was found to be $\langle F \rangle = 3.2 \pm 0.3$ pN (standard deviation). This value was determined from a histogram of all the pixel-wise values, as shown in the tension map in Fig. 3C. Moreover, when a bimodal fitting model is applied to the force histogram, two distinct peak forces are identified, suggesting the presence of two sub-populations: $x_1 = 3.1 \pm 0.4$ pN and $x_2 = 3.5 \pm 0.1$ pN (Fig. 3C and Table 3).

## Discussion

In this study, we developed and validated a three-colour Cascade-FRET system capable of quantifying complex molecular interactions and dynamics in vitro and in cells. Our methodology expands pair-wise FRET by

**Fig. 3 | Three-colour FRET model applied to the vinculin tension-sensing biosensor with mScarlet-RIAM in fixed MEF$^{\text{Vinc-/-}}$ cells. A** Structural representations of the VincTS biosensor in complex with mScarlet–RIAM (1–27 aa). The left panel shows a schematic model created in BioRender®, and the right panel shows a molecular model generated in PyMOL using PDB structures (7PNN, mVenus; 4Q9W, mTurquoise2; 5LK4, mScarlet-I) and AlphaFold3 predictions of vinculin, RIAM, and the spider-silk linker (GPGGA)$_8$. **B** Fluorescence lifetime data for VincTS with and without mScarlet–RIAM. Representative FLIM and epifluorescence images are shown on the right. Scale bar = [insert μm if known]. **C** Average corrected force per cell for MEF$^{\text{Vinc-/-}}$ cells transfected with VincTS ± mScarlet–RIAM. Each data point represents the mean force per cell; bars indicate the pooled population mean. Right panels show corrected lifetime and force maps of a representative VincTS + mScarlet–RIAM cell, with corresponding lifetime and force distributions below. Red = Gaussian fit; blue = first component of bimodal fit; black = second component. Statistical analysis was performed by one-way ANOVA with Tukey's correction for multiple comparisons: $P \geq 0.123$ ns (ns), $P \leq 0.0332$ (*), $P \leq 0.0021$ (**), $P \leq 0.0002$ (***), $P \leq 0.0001$ (****).

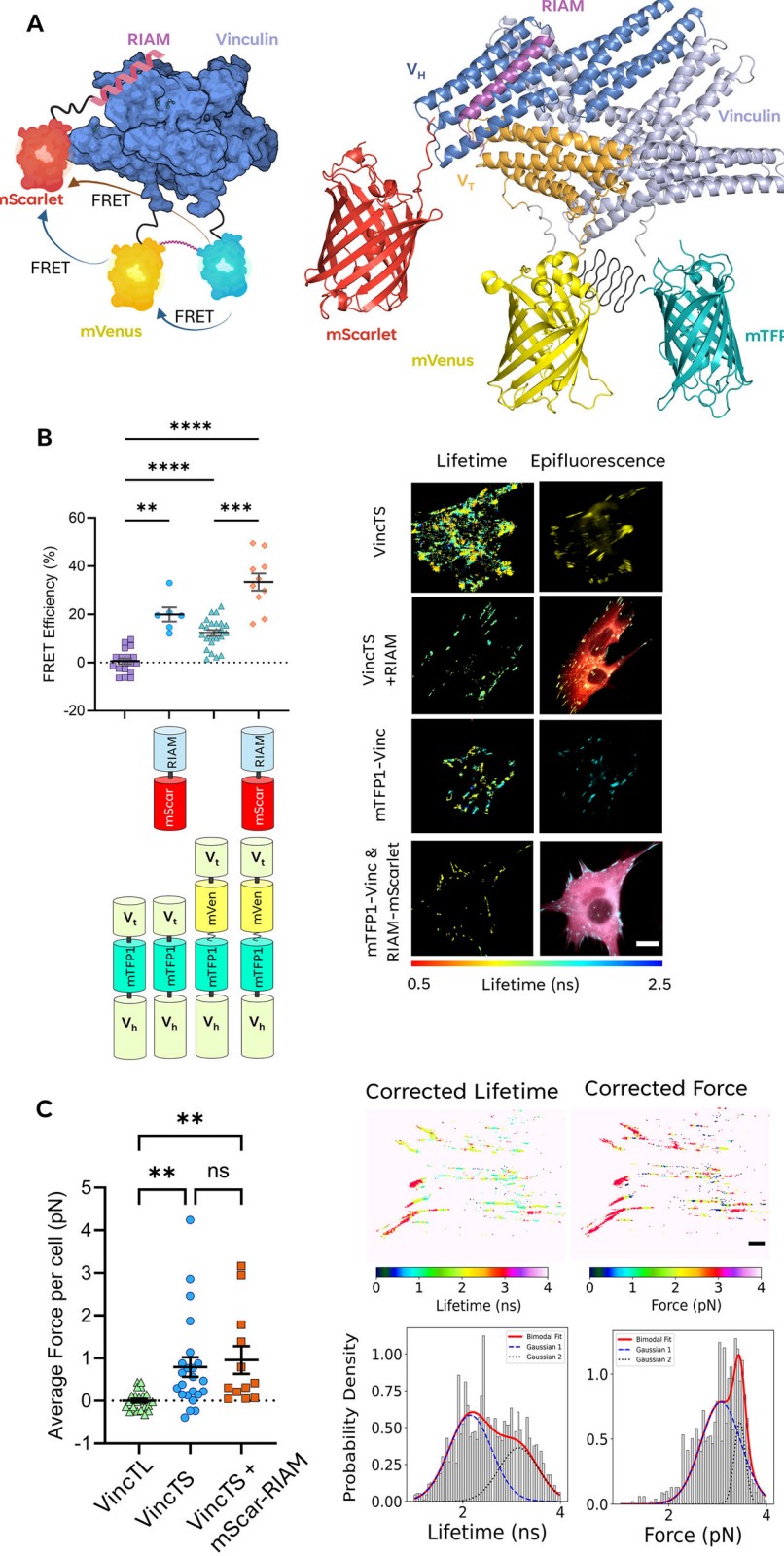

integrating an additional energy transfer step, providing a more nuanced approach for elucidating the spatial organisation of biomolecular complexes. We applied this system to study the interaction of vinculin, a mechanosensitive focal adhesion protein, with its putative binding partner, RIAM, under tension.

The results obtained from purified protein constructs confirm the predictive capabilities of the FRET-cascade system, as these results recapitulate an Alphafold3-predicted structure. Discrepancies in the predicted separation distances are accounted for in the experimental error and likely structure relaxation due to solvation[46]. The results demonstrate that measurements of FRET transition rates can be used to verify structures and distances using a robust and straightforward theoretical framework. This represents a novel approach to the analysis of structure-function relationships.

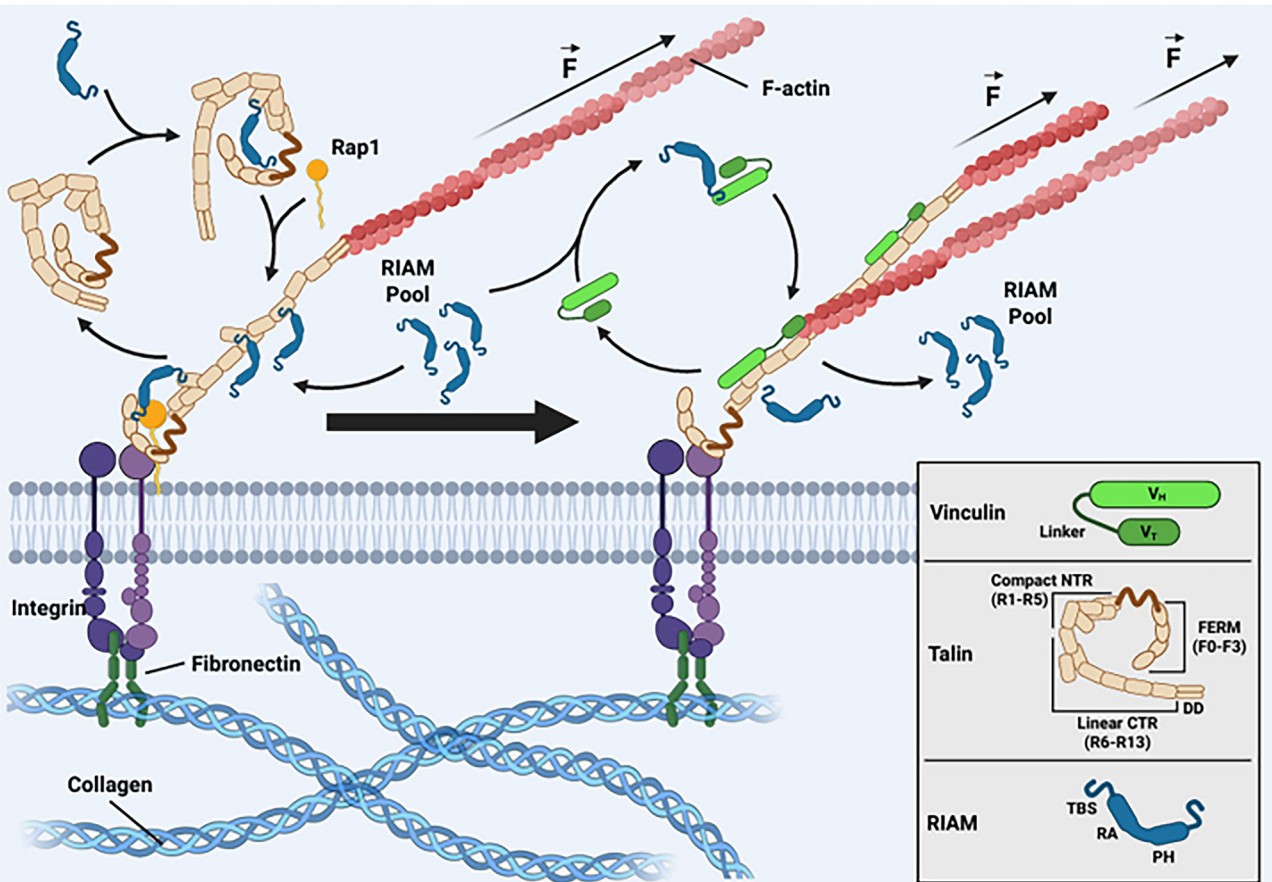

**Fig. 4 | A schematic detailing the key events in our proposed model of vinculin recruitment by RIAM.** This scheme illustrates the interaction between the vinculin and RIAM and the activation of vinculin through the binding of talin and f-actin. It also demonstrates how the proposed recruitment of vinculin and talin by RIAM is expected to work. $\vec{F}$ = Tensile force; NTR = N-terminal region; CTR = C-terminal region; DD = dimerisation domain. Thin arrows represent the movement of proteins; the thick block arrow represents the evolution of the FA maturation process. The figure was made using BioRender®.

We suggest that FRET measurements can be used to determine unknown structures which correlate with a predictive model (Alphafold3). Distances between FRET pairs were determined using FRET efficiencies and Förster radii (Supplementary Tables 1 & 2). A model of the three-colour mTurq2-mVenus-mScarlet-I protein predicted shorter distances between fluorophores compared to those measured from the energy transfer rates and determined by negative stain TEM (Supplementary Table S2). The separation distances are all in general agreement, describing a trigonal planar arrangement for the three-colour FRET cascade model protein. To fully determine FRET transfer rates within the structure, we introduced a novel G68A mutation in mVenus to produce a non-fluorescent and non-absorbing variant as a FRET control without significantly altering the structure of mVenus (Supplementary Fig. S4). This is essential for the characterisation of the model system. While other mutations could have been used to create a dark version of mVenus, such as the Q69M and F46L[60,61], they do not prevent the formation of a functional fluorochrome in the same manner as the G68A mutation.

We successfully demonstrated that vinculin interacts with RIAM in cells, a finding that corroborates previous in vitro studies[38]. Our results provide evidence for the direct binding of RIAM to the N-terminus of vinculin, with higher FRET efficiencies observed for N-terminally tagged RIAM compared to its C-terminally tagged counterpart. This agrees with AlphaFold3 predictions, underscoring the importance of the N-terminus in the interaction.

Using a vinculin tension sensor, we quantified the forces exerted on vinculin within FA. The average force, ⟨F⟩, determined by the average FRET efficiency and Eq. 5, was 3.0 ± 0.3 pN, which is consistent with previous reports[36] and underscores the heterogeneity of tension in individual adhesions. This heterogeneity likely reflects the dynamic nature of FA, where vinculin undergoes rapid changes in positioning, F-actin association and binding partners in response to mechanical cues[62–65]. FRET-cascade allowed us to spatially resolve these forces, revealing how the interaction of vinculin with RIAM correlates with changes in mechanical tension. The average force across vincTS in the presence of mScarlet-RIAM, when corrected for the energy transfer between mTFP1 and mScarlet, was found to be 3.2 ± 0.3 pN. This value is within the standard error of the vincTS alone, where ⟨F⟩ = 3.0 ± 0.3 pN (Fig. 3C). Furthermore, when comparing the average tensile forces across each cell for the vincTL, vincTS, and vincTS + mScarlet-RIAM conditions, we report that there is no significant difference in average cellular force between the two vincTS conditions (Fig. 3C). This indicates that the vinculin-RIAM complex associates when a moderate force is acting on vinculin. Such insights are crucial for understanding how FA adapt to mechanical stress and how vinculin acts as a molecular clutch to transduce forces between the ECM and the cytoskeleton[49,65–67].

Our results indicate that RIAM is associated with the auto-inhibited conformation of vinculin, in a manner analogous to RIAM binding to auto-inhibited talin[38], and that this complex can relocate to the plasma membrane (Fig. 4). Interestingly, we observe relatively consistent RIAM–vinculin association in the cytoplasm, but higher variance at FA (Fig. S13), potentially reflecting a transient persistence of the RIAM–vinculin complex after vinculin engages talin. Importantly, our findings remain consistent with the established view that RIAM–talin and vinculin–talin interactions are

mutually exclusive[38]. Rather, we propose a model in which RIAM first engages vinculin under low-tension conditions, consistent with our observation that mScarlet-RIAM binds to VincTS when the construct reports a high average FRET (low tension) (Fig. 3b). The absence of a detectable FRET changes between GFP-vinculin and mScarlet-RIAM upon ROCK inhibition (H-1152) further suggests that RIAM–vinculin binding is not dependent on actomyosin-generated forces (Supplementary Fig. S17). We therefore suggest that RIAM may transiently bind vinculin before dissociating and subsequently engaging talin, where force-dependent[49,59] remodelling of talin enables vinculin binding to the R2/3 domains of talin[49]. Our evidence of RIAM–vinculin interaction in FAs is consistent with RIAM acting as a mediator of vinculin recruitment and translocation, without challenging the notion that vinculin and RIAM cannot simultaneously occupy their respective talin-binding sites.

The FRET-cascade approach offers several advantages over pair-wise FRET methods, including the ability to dissect complex, multi-component interactions with high precision. This methodology can be readily adapted to study other dynamic protein complexes, such as those involved in signal transduction, cytoskeletal organisation, or transcriptional regulation.

Future studies could expand the system to include additional fluorophores, enabling the exploration of even more complex biological networks. Moreover, integrating FRET-cascade with advanced imaging techniques, such as super-resolution microscopy, could provide unprecedented insights into molecular interactions at the nanoscale. Improvements in computational modelling, such as incorporating Amber relaxation into structural predictions, could further improve the agreement of the AlphaFold model with FRET and TEM-based distance measurements.

In summary, our three-colour FRET-cascade system provides a powerful tool for investigating dynamic molecular interactions in vitro and in cells. By applying this methodology to vinculin and RIAM, we have uncovered new insights into the molecular mechanisms of focal adhesion assembly and mechanotransduction. These findings validate the FRET-cascade system and highlight its potential for broad applications in cell biology and beyond.

## Methods

### Cell culture & Lipofectamine 3000 Transfection
Vinculin null, Mouse Embryonic Fibroblasts (MEFs) were cultured in high glucose DMEM (Merck, D6429) supplemented with 10% Foetal Bovine Serum (FBS) (Gibco, A4766801), 1% Penicillin-Streptomycin (Thermo Fisher, 15140122), 2 mM L-Glutamine (Thermo Fisher, 25030081) and 1x MEM non-essential amino acids (Thermo Fisher, 11140050), referred to hereafter as complete growth culture medium. Cells were incubated at 37 °C with 5% $CO_2$. MEFs were seeded at 20,000 cells per 35 mm, on fibronectin-coated glass-bottomed μDish (Ibidi, 81158). Plasmid DNA was transiently transfected using Lipofectamine 3000 in a 2:1 DNA:Lipofectamine ratio, (Thermo Fisher, L3000001) per dish.

### Cloning & Site-Directed Mutagenesis
The three-colour FRET mTurq2-mVenus-mScarlet-I construct was initially designed using SnapGene® and was made by VectorBuilder® on a proprietary mammalian expression vector. The different FP pairs and single FP constructs were constructed through a series of digestions with restriction endonucleases, which were used to excise specific FPs from the parent three-colour construct. The resulting linearised DNAs were then ligated together with T4 DNA ligase and transformed into chemically competent E. coli DH5-α cells (Thermo Fisher, 18258012) before plasmid purification using a HiSpeed® ® Plasmid Midi purification kit (Qiagen, 12643). A separate set of constructs were required for bacterial expression. This was achieved by PCR amplification of the required FPs, which were first gel-purified and then ligated into a pET151 directional TOPO™ Expression system. Site-directed mutagenesis (SDM) introduced a single point mutation into the mVenus fluorescent protein of the three-colour construct at amino acid 67, glycine, which was mutated to an alanine. A pair of primers was designed containing a two-base-pair mismatch in the centre of the primer pair. A PCR reaction was then carried out using the Q5® High-Fidelity DNA polymerase (NEB, M0492L), which produced a linearised form of the double-stranded parental template. The linear plasmid was treated with KLD (Kinase, Ligase, and DpnI) mix as part of the SDM kit (NEB, E0554S).

### Protein purification
The fluorescent protein constructs were expressed in the E. coli BL21 (DE3) strain, where single colonies were picked and grown overnight at 37 °C in 10 mL of LB media before they were used to inoculate 250 mL of auto-induction media[68] and grown at 18 °C for 72 h. Cells were harvested by centrifugation at 10,000 g for 20 min at 4 °C and resuspended in lysis buffer: 50 mM Tris-Cl adjusted to pH 8.0, with 150 mM NaCl, 20 mM Imidazole and protease inhibitor cocktail (Roche) and 5 units/mL of Benzonase endonuclease (Merck). Cell Lysis was achieved by sonication (30% power for a total sonication time of 1 min and 30 s) using a Sonics Vibra-cell VC 750 sonicator. The resulting cell suspension was centrifuged at 18,750 g for 45 min at 4 °C. The supernatant was removed and micro-filtered through a 0.22 μm pore disc filter (Millipore) before loading on a 1 mL HisTrap column (GE Healthcare) pre-equilibrated with lysis buffer for immobilised metal-ion affinity chromatography (IMAC). FPs were eluted using an imidazole gradient run on an Äkta Pure FPLC chromatography system.

Fractions containing the FPs of interest were collected, pooled, and dialysed overnight at 4 °C in an imidazole-free lysis buffer. The Hexa-Histidine purification tag was cleaved by incubating the dialysed sample for approximately 8 h at room temperature in the presence of the Tobacco Etch Virus (TEV) protease. Once the tag was removed, the fluorescent protein samples were again filtered through a 0.22 μm pore disc filter before re-loading onto the same HisTrap column (GE Healthcare), which was again pre-equilibrated with lysis buffer. The untagged proteins flowed through the column at a flow rate of 5 mL/min and were then collected in the flow-through. The untagged proteins were concentrated using a Pierce™ 10 K MWCO Protein Concentrator before size exclusion chromatography (SEC) on a 16/60 HiLoad Superdex 75 column (GE Healthcare) equilibrated with 50 mM Tris-HCl, pH 8.0, and 150 mM NaCl adjusted to pH 8.0.

### Immunoprecipitation by GFP-Trap and Western Blotting
MEF$^{Vinc-/-}$ cells were seeded at $2 \times 10^6$ per 100 mm dish and transfected 24 h later with either GFP-only or GFP–vinculin plasmids using Lipofectamine 3000 (Thermo Fisher) according to the manufacturer's instructions. One dish per condition was treated with 20 μM nocodazole (Abcam) for 30 min prior to lysis. Cells were washed twice with ice-cold PBS and then lysed in 750 μL of lysis buffer (50 mM Tris-Cl, 150 mM NaCl, 500 μM EDTA, 0.1% IGEPAL, 25 mM NaF, 0.5% Triton X-100, and 1 mM PMSF). Lysates were mechanically disrupted by passage through a 25 G needle, incubated on ice for 30 min, and then centrifuged at $17,100 \times g$ for 10 min at 4 °C. Cleared lysates were kept on ice; 20 μL was reserved as total input and mixed with 2× Laemmli buffer.

For GFP pull-down, 25 μL of GFP-Trap® Magnetic Agarose beads (ChromoTek) were pre-washed and incubated with 500 μL of lysate and 500 μL of dilution buffer (50 mM Tris-Cl, 150 mM NaCl, 500 μM EDTA, 0.1% IGEPAL, 25 mM NaF, and 1 mM PMSF) for 2 h at 4 °C on a rotator. Beads were washed, eluted in 2× Laemmli buffer, and boiled for 10 min. Proteins were resolved using 3–8% Tris-Acetate or 10% Bis-Tris NuPAGE gels (Invitrogen) and transferred to 0.45 μm nitrocellulose membranes at 100 V for 2 h at 4 °C. Membranes were blocked in 5% (w/v) milk in TBS-T for 1 h, incubated overnight at 4 °C with primary antibodies [Mouse anti-Talin1 (8d4 monoclonal, Merck), or Rabbit anti-RIAM (EPR2806 monoclonal, Abcam)], and then with secondary antibodies [Goat anti-Mouse IRDye® 680RD, LI-COR; or Goat anti-Rabbit IRDye® 800CW, LI-COR] for 1 h at room temperature. Blots were imaged using the Odyssey CLX system (LI-COR).

### Multiphoton TCSPC FLIM and Analysis
Transfected cells were cultured on borosilicate glass coverslips (VWR, Thickness No. 1.5) coated with fibronectin and fixed in 4% PFA-PHEM

**Article**

solution for 30 min at room temperature, 48 h post-transfection. Subsequently, the cells were permeabilised with 0.2% Triton-X in 50 mM Tris Buffered Saline (TBS) for 20 min at room temperature and then quenched with 1 mg/mL $NaBH_4$ (also in 50 mM TBS) for a further 20 min at room temperature. The Multiphoton-FLIM TCSPC imaging system is a custom system constructed around a Nikon Eclipse Ti-E microscope. This was fitted with a 40 × 1.30 NA Nikon Plan-Fluor oil objective and an 80 MHz Ti:Sapphire laser (Chameleon Vision II, Coherent) tuned to 875 or 950 nm for two-photon excitation of mTurquoise2 or mVenus, respectively. Photons were collected using a 480/30 nm emission filter for mTurquoise2 or a 525/25 nm emission filter for mVenus (all from SemrockTM) and an HPM 100-40 hybrid detector (Becker & Hickl GmbH). Laser power was adjusted to give average photon counting rates of $10^4$ to $10^5$ photons $s^{-1}$, with peak rates approaching $10^6$ photons $s^{-1}$. Acquisition times of 300 seconds at low excitation power were used to achieve sufficient photon statistics for fitting while avoiding either pulse pile-up[3,69] or significant photobleaching. All FLIM data were analysed using a time-resolved image analysis package, TRI2[6], and were fitted with either a mono-exponential or biexponential decay curve using the Levenberg-Marquardt algorithm. Lifetime data processed in TRI2 produced histograms of pixel frequencies against photon arrival times for every FP imaged in each cell condition and experiment. A custom Python script was written, which imported these histograms for each cell in each experiment into a single data frame, and an intensity-weighted average lifetime was calculated for each cell imaged. An average of 10–15 cells per cell condition was imaged in each experiment, and an unweighted average of these (each cell imaged given an equal weighting) was calculated. Mean lifetimes were used to calculate FRET efficiencies and energy transfer rates within the same Python script. Graphs illustrating the spread of average lifetimes and FRET efficiencies, along with all corresponding statistical tests, were generated using GraphPad Prism 9.

### Domain-anchored Cα superposition

Two tri-β-barrel constructs were analysed: an mTurquoise2–mVenus–mScarlet-I construct and an mTFP1–mVenus–mScarlet-I construct. For mTurquoise2, the barrel ranges were 1–238 (mTurq2), 249–487 (middle, mVenus), and 498–731 (mScarlet-I). For mTFP1, the ranges were 1–235 (mTFP1), 246–484 (middle, mVenus), and 495–726 (mScarlet). For each sequence, five AlphaFold3 models were generated from different random seeds. Within each five-model set, structures were superposed on the middle mVenus barrel using Cα atoms, restricting the fit to the middle-barrel residue range for that construct. After this superposition, Cα root-mean-square deviations (RMSDs) were calculated separately for the N, M middle, and C barrels to characterise rigid-body motion of the outer barrels arising from the flexible linkers. To summarise the tri-barrel geometry, a pseudoatom was placed at the centroid of each barrel. Then, the distances |N–M| and | M–C |, along with the opening angle ∠N–M–C at the middle centroid, were recorded for each model. Mean and standard deviation were reported across the five seeds. Between constructs, the isolated N-terminal barrels were compared using Combinatorial Extension (CEalign in PyMOL®) to provide a sequence-independent fold comparison, with the CE RMSD and aligned length reported. For full-construct comparison, a representative model was chosen per set by the medoid criterion on the middle-barrel RMSD. One construct was placed into the coordinate frame of the other by superposing on the middle barrel, and geometric measurements were repeated in that common frame.

### Force mapping and visualisation

A Python script was written to convert lifetime measurements into force measurements within FA. The histogram of lifetime data per pixel was loaded into the script, and a mathematical transformation was applied to the lifetime values, resulting in force values. The processed data is then reshaped into a structured format and converted into images—one representing the distribution of forces and another showing molecular lifetimes.

## Data availability

All data supporting the findings of this study are available from the corresponding author upon reasonable request. Prism files have been uploaded to Figshare and can be found here https://doi.org/10.6084/m9.figshare.30642743.

## Code availability

Custom Python scripts used in this study are available from the corresponding author upon reasonable request.

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

## Acknowledgements
The authors thank Ambrish Kumar for their assistance with protein purification, Mark Pfuhl for providing TEV protease, purification and the Nikon Imaging Centre @King's staff for assistance with spinning disc confocal imaging. We acknowledge funding from the UK MRC MR/X012794/1 and Cancer Research UK for Simon M. Ameer-Beg and MRC/EPSRC for the doctoral scholarship of Conor A. Treacy.

## Author contributions
C.T., M.P., M.A.P., and S.A.B. designed the research. C.T., S.P., and T.B. performed research. C.T., T.L.P., M.P., T.B., and S.A.B. wrote the paper.

## Competing interests
There are no competing interests.
