## [Transparent Peer Review file · Communications Chemistry]

Development of three-colour cascade FRET for force sensing of the putative RIAM-Vinculin interaction in fibroblasts

Corresponding Author: Professor Simon Ameer-Beg

Version 0:

Reviewer comments:

Reviewer #1

(Remarks to the Author)

Focal adhesions are essential structures used by adherent cells to anchor themselves to their surroundings and transmit mechanical signals. In this study, Treacy et al. present a biosensor designed to measure force changes within vinculin, a key component of focal adhesions. The biosensor operates through fluorescence lifetime changes in response to mechanical force across vinculin, enabling real-time visualization of intracellular tension during focal adhesion development.

A major finding of this work is the force-independent interaction between RIAM, a protein involved in adhesion signaling, and vinculin. This was elucidated using a novel three-color Förster resonance energy transfer (FRET) system that tracks the dynamic interplay between talin, vinculin, and RIAM. This approach provides valuable insight into the temporal order and mechanism of focal adhesion assembly.

The authors developed a three-color FRET system comprising the fluorescent proteins mTurquoise2, mVenus, and mScarlet-I, linked via flexible GGSGGS peptide linkers. The system was rigorously validated through a combination of purified protein studies, spectroscopic analysis, structural modeling, and negative-staining transmission electron microscopy (TEM). FRET measurements were performed in live cells using time-correlated single-photon counting fluorescence lifetime imaging microscopy (TCSPC-FLIM), a widely accessible and robust technique.

RIAM–vinculin interactions were further characterized in vinculin-knockout mouse embryonic fibroblasts. The data clearly demonstrate that RIAM binds to the N-terminus of vinculin at focal adhesions and that this interaction is dependent on an intact microtubule network. Using vinculin-based tension sensors, the authors measured an average intracellular force of 3.0 ± 0.3 pN per focal adhesion, reinforcing vinculin's mechanosensitive role while confirming that its interaction with RIAM occurs independently of mechanical tension.

This study highlights the utility of combining three-color FRET with TCSPC-FLIM to investigate the composition and mechanics of dynamic, multicomponent molecular complexes in living cells. The mechanistic insights into RIAM–vinculin interactions and their modulation by cytoskeletal organization represent a significant contribution to our understanding of cell adhesion.

I particularly appreciate the elegant application of fluorescence lifetime imaging for intracellular force measurement. This technique enables precise spatial and temporal resolution, making it a powerful tool at the interface of cell biology and mechanochemistry.

The manuscript is exceptionally well-executed and clearly presented, particularly given the technical complexity of the methodology. The main findings are compelling and well-supported. My comments are primarily concerned with presentation and clarity:

Minor Points

Orientation Factor in R_0 Calculations

The orientation factor (κ^2) plays a critical role in determining FRET efficiency and, by extension, molecular distances. The manuscript does not specify the assumed value. Given the use of flexible six-amino-acid GGSGGS linkers and the evidence of considerable fluorophore mobility shown in Figure S5, a value of $\kappa^2 = 2/3$ (random orientation) seems appropriate. While the FRET-derived distances correlate well with TEM data (Figure S9), it would strengthen the methods section to explicitly state the orientation factor used in R_0 calculations.

Lifetime Color Palettes in Figure 3

The fluorescence lifetime color palettes in Figure 3 appear inconsistent. In particular, panels 3b and 3c use different palette

scales within the same figure. This may be confusing for readers and should be standardized to improve clarity.

Typographical Error on Page 19

The phrase "Supplementary tables" is unnecessarily repeated on page 19 and should be corrected.

Reviewer #2

(Remarks to the Author)

The manuscript submitted by Treacy et al. describes the use of a multistep Förster resonance energy transfer (FRET) system to visualise the interaction between RIAM and vinculin. This multistep process involves three colours and enables the potential force dependency in RIAM-vinculin interactions to be studied. In this process, vinculin tension sensor modules contain two fluorophores that detect mechanical forces acting on vinculin and RIAM. These are linked to a third fluorophore that detects the interaction with vinculin. While the molecular and cellular aspects are not particularly novel, as it has previously been demonstrated that vinculin binds to RIAM (PMID: 23389036), this study shows how Cascade-FRET could elucidate molecular interactions and dependencies using activation sensors. Unfortunately, the authors use the vinculin-RIAM pair to test this system. This makes limited sense if proteins bind irrespective of their activation state (as seems to be the case here). Nevertheless, the study may have technical value if the authors include essential controls and address the points below.

1) Experiments related to Fig. 2c require an additional negative control using a focal adhesion protein that does not bind to RIAM, such as Paxillin-GFP. It is anticipated that paxillin-GFP will not undergo FRET when co-expressed with RIAM-mScarlet. Likewise, experiments involving the tension sensor (Fig. 3b) would benefit from additional control constructs, such as paxillin-mTFP and paxillin-mTFP-mVenus.

These experiments would demonstrate that molecular crowding in focal adhesions does not contribute to FRET, while specific interactions between vinculin and RIAM do.

2) Figure 2(d) is difficult to understand. Why is the FRET efficiency similar (not significant) for VincTL and VincTS? Wouldn't you expect to see a significant difference in the blots presented on the left-hand side of the figure? Could the force distribution maps below the VincTL and VincTS FRET distributions be mixed up? I would have expected to see low force (high FRET) in the VincTL sample and higher force (low or heterogeneous FRET) in the VincTS sample. However, this would contradict the non-significant FRET efficiency values on the left.

3) Only lifetime images of focal adhesions are shown, but the schematic model also shows RIAM binding in the cytoplasm. It would be interesting to determine whether vinculin binds to RIAM in the cytoplasm. Therefore, it would be beneficial to present FRET values for areas outside of focal adhesions in a separate figure (see also comment below).

4) The model does not reflect current knowledge about RIAM. According to this knowledge, Rap1 first recruits talin at the membrane, forming a ternary complex with RIAM. Furthermore, the model presented here does not consider that the RIAM TBS site and the talin-vinculin binding site (VBS) bind to the same region of vinculin. This precludes the possibility of RIAM binding to both talin and vinculin simultaneously, as shown in the model figure. Instead, the model suggests that the main RIAM-vinculin interaction site may be in the cytoplasm, which could prevent vinculin from binding to talin. If vinculin can be recruited to focal adhesions by other proteins without binding to talin, one might imagine direct vinculin-RIAM binding also in focal adhesions.

Minor points:

- 1) Fig 2 figure legend had the title duplicated.
- 2) Several of the reference numbers appear to be out of line with the in-text citations.
- 3) The title does not make sense if RIAM binds vinculin in a force-independent manner.

Reviewer #3

(Remarks to the Author)

In this study, the authors introduce a three-color FRET cascade method and its application in detecting the binding of vinculin and RIAM within cells. In the manuscript, the aim is clear: to develop a methodology and how the authors want to apply it. However, it is concerning that the method described in Fig. 1 and the cellular experiments in Figs 2 and 3 employed different fluorescent proteins, raising questions about the consistency and validity of the study's design. The authors need to revise this inconsistency before this study can be published. I list my comments below for points that would improve this manuscript.

Major points:

1. It is inappropriate that the authors used mTurq2 in Fig. 1, while mTFP1 was used for FRET measurements in Figs. 2 and 3. The same fluorescent protein, mTurq2, should be used to demonstrate the development of the three-color FRET method and its application. In addition, the mVenusG68A mutant used in Fig. 1 was not used in the cellular experiments in Figs. 2 and 3.
2. In Fig. 2c and d, Fig. 3b and c, measuring FRET efficiency and force over the entire cell yields poor spatial resolution. Within a single cell, force exerted on focal adhesions is heterogeneous, and even within individual focal adhesions, force distribution can vary. Averaging across the whole cell obscures these local variations in force and FRET signals. The

analysis should instead be restricted to subcellular regions, particularly focal adhesions.

3. The Materials and Methods section does not provide sufficient information. The description of a western blot of the co-IP of EGFP-vinculin with RIAM and Talin (Fig. 2b) is lacking. No detail of the vinculin^{TL} and vinculin^{TS} constructs were given.

Minor points:

4. The reference numbers given in the main text are incorrect (I noticed this around Ref. 38, p2).

5. Fig. 2b, right panel, statistical significance should be shown.

Version 1:

Reviewer comments:

Reviewer #1

(Remarks to the Author)

Thank you for the careful and constructive revision. I have read your point-by-point response and re-checked the amended manuscript.

The typographical errors have been corrected.

The harmonised FLIM colour look-up tables in Figures 2–3 resolve the earlier palette inconsistency; the figures now read clearly.

The Methods now explicitly state that $\kappa^2 = 2/3$ is assumed (lines 134–136), which addresses my concern about the orientation factor. I appreciate your rationale for keeping a deeper treatment (e.g., time-resolved anisotropy or single-molecule orientation measurements) out of scope for this paper.

I am satisfied with the changes, from my point of view the manuscript is clear and ready for publication.

Reviewer #2

(Remarks to the Author)

I am happy with the changes made. Whilst the probe does not reveal much novelty, experiments using such probes may give some technical insight to the reader community.

Reviewer #3

(Remarks to the Author)

The authors satisfactorily responded to my comments.

Dear Editor,

We would like to thank you and the reviewers for the time and effort invested in evaluating our manuscript, "*Development of three-colour cascade FRET for force sensing of the putative RIAM-Vinculin interaction in fibroblasts*". We greatly appreciate the constructive comments and suggestions from the reviewers, which have helped us to improve the clarity and robustness of the work. We note that reviewer 1 had no significant concerns regarding the work and was broadly supportive, whilst reviewers 2 and 3 had more significant conceptual issues; however, we would note that none of these constitute major revisions. We have carefully considered each point raised by the reviewers and have revised the manuscript accordingly. We have added line numbers to the manuscript to improve the clarity of our revisions. Taking each reviewer in turn:

Reviewer 1:

The typographical error on page 19 (repeated phrase) has been corrected. In agreement with the reviewers, we have harmonised the fluorescence lifetime colour palettes across all figures. Colour look-up tables are now consistent throughout Figures 2 and 3. Regarding the reviewers' pertinent remarks about the orientation factor used in FRET calculations, we confirm that $\kappa^2 = 2/3$ is assumed throughout the manuscript. Whilst we have considered further investigation of the role of orientation in this sensing model ~ (particularly in light of the short linker lengths), elucidation would require significant investment in spectroscopic measurements of time-resolved anisotropy or single molecule measurements for surface bound proteins (comparable to the TEM data in Supplementary Figure S10) which, we feel, are beyond the scope of this manuscript and would likely form the basis of an additional publication. New sentences have been added at lines 134 and 136. We thank Reviewer 1 for concise and practical guidance, all of which has been addressed in our revision.

Reviewer 2:

To address the comments from reviewer 2 regarding additional negative controls, we have conducted additional control experiments to demonstrate that FRET does not occur serendipitously between mScarlet-RIAM and Paxillin-eGFP, which is also expressed in focal adhesions. As expected, no lifetime reduction is observed when these two constructs are co-expressed, and therefore, no FRET occurs. This is consistent, given there is no reported association between paxillin and RIAM in the literature, despite their colocalization within focal adhesions. Considering this negative result, we feel that the additional experiment requested to demonstrate the same negative result for Paxillin-mTFP1 and a Vinculin-mTFP1-mVenus would not result in any additional clarity for the reviewer given paxillin and RIAM proteins demonstrably do not demonstrate FRET for the most likely serendipitous interaction. We thank the reviewer for their careful inspection of Figure 2(d) which they correctly found confusing. We identified that the labelling of the figure was mismatched and have now revised this figure accordingly to address this confusion. The reviewer correctly notes that the data presented in Figure 2(c) are thresholded to predominantly show the focal adhesions. We have now provided a supplementary figure, Figure S13 which now includes the segmented focal adhesions, whole cell and cytoplasm (FA excluded) only data for the same data shown in Figure 2. Figure S13 clearly demonstrates that the interaction between Vinculin and RIAM occurs throughout the cell supporting our assertion presented in the model Figure 4. In regard to the model presented in Figure

4, we thank the reviewer for raising concerns about the mechanism presented and have adjusted components of the figure to avoid confusion and better align with current knowledge of RIAM in focal adhesion formation. RIAM plays a role in the recruitment of Talin in certain contexts but is dispensable in others (Goult et al., 2013; Bromberger et al., 2019), but in any case, Rap1 can interact with the FERM domain of talin (Bromberger et al., 2019) and we have modified the model figure to reflect this. We are unsure what the reviewer means by “the RIAM TBS site and the talin-vinculin binding site (VBS) bind to the same region of vinculin” and assumed the reviewer is saying that the RIAM binding site and Vinculin binding site *on talin* are overlapping and therefore RIAM and vinculin binding to talin are mutually exclusive (Goult et al., 2013) If this is so, then we agree with the reviewer and have adjusted the model, we wanted to clarify that we never intended to challenge this notion and therefore reiterated this in the text as part of a re-written discussion (lines 325-338). That said, this notion is no longer so clear cut, and at the R11 VBS site of talin, the binding of vinculin and RIAM are independent and non-overlapping (Vigouroux et al., 2020). The reviewer draws a correct implication from the model that RIAM and vinculin may interact in the cytoplasm, and this is supported by the new data presented in Fig. S13, which shows that vinculin and RIAM interact in the cytoplasm consistently, whereas there is much higher variance in the focal adhesions, potentially indicating that RIAM and vinculin remain associated for at least a measurable time after vinculin associates with talin but some are dissociated. The reviewer correctly points out that the overlapping binding sites of RIAM and talin within vinculin would result in RIAM sterically hindering vinculin binding to talin, but whether a cytoplasmic RIAM-vinculin interaction prevents vinculin binding to talin likely comes down to differential binding affinities. Given that previous research demonstrated that vinculin has a significantly lower affinity for RIAM compared to talin-R3-VBS (Goult et al., 2013) it is likely the displacement of RIAM for talin on vinculin is driven simply by binding kinetics (given the force-driven exposure of the talin-VBS sites has already occurred).

We again thank the reviewer for careful reading of the manuscript and have corrected typographical errors (repetition of Fig2 legend) and corrected the issue with in-text citations. After some consideration we agree with the reviewer and have updated the title to reflect the force independent manner of RIAM-Vinculin binding

Reviewer 3's comments were addressed by providing a detailed rationale for our fluorophore choices (mTurquoise2 versus mTFP1), clarifying why it was not appropriate to alter the established vinculin tension sensor, and demonstrating how our approach ensured both methodological innovation and biological validity. We have also added supplementary analyses of focal adhesion versus cytoplasmic regions, expanded the Materials and Methods section to address missing information, corrected EndNote-related reference issues, updated the discussion and schematic model to reflect current knowledge on RIAM–vinculin–talin interactions, and added statistical significance to Figure 2b. We also updated the title to reflect the conclusion that RIAM–vinculin binding is force-independent.

Reviewer 3:

We thank reviewer 3 for their insightful comments regarding fluorophore choice in our model system and then subsequent biological exemplification with a similar but spectroscopically different fluorescent protein. However, we respectfully disagree that it is necessary to repeat the biological exemplification with mTurquoise2.

We agree that consistency is desirable. The VincTS biosensor was designed with mTFP1 and published in *Nature Methods* in 2010 (Grashoff, C. et al. Measuring mechanical tension across vinculin reveals regulation of focal adhesion dynamics. *Nature* 466, 263-266 (2010)), Brenner, M. D. et al. Spider Silk Peptide Is a Compact, Linear Nanospring Ideal for Intracellular Tension Sensing. *Nano Lett* 16, 2096-2102 (2016) and Becker, N. et al. Molecular nanosprings in spider capture-silk threads. *Nat Mater* 2, 278-283 (2003). <https://doi.org/10.1038/nmat858>). It has been thoroughly validated and is widely used. We therefore maintain mTFP1 as the donor in all VincTS measurements, which preserves continuity with the established literature and allows us to use the validated stiffness values for cellular force calculations.

Supplementary Figure S7 is a conceptual schematic of the three-colour cascade. To assess whether donor choice could affect the mechanism, we performed a structure-based analysis using five AlphaFold3 models per construct and a domain-anchored C α superposition on the middle mVenus barrel. The geometries are similar and FRET-compatible for both designs. Across seeds:

- **mTurquoise2 construct:** mTurq2→mVenus 37.5–44.4 Å, mVenus→mScarlet 35.8–42.8 Å, angle 49.6–69.2°, means \pm SD 40.75 \pm 3.19 Å, 38.88 \pm 2.88 Å, 59.95 \pm 8.36°.
- **mTFP1 construct:** mTurq2→mVenus 43.2–53.0 Å, mVenus→mScarlet 33.2–36.6 Å, angle 36.6–47.2°, means \pm SD 48.20 \pm 4.06 Å, 34.49 \pm 1.32 Å, 42.32 \pm 4.66°.

The first-barrel folds are the same by Combinatorial Extension alignment (CEalign) (Shindyalov, I. N., and Bourne, P. E. 1998. "Protein structure alignment by incremental combinatorial extension (CE) of the optimal path." *Protein Engineering* 11(9): 739–747. doi:10.1093/protein/11.9.739.), RMSD 2.61 Å over 224 residues. Thus, the cascade geometry does not hinge on whether the N-terminal donor is mTFP1 or mTurquoise2.

Time-resolved readouts are measured in terms of donor lifetimes we have measured lifetimes of ~2.22 ns for mTFP1 and ~4.15 ns for mTurquoise2, consistent with published lifetimes (Goedhart, J., von Stetten, D., Noirclerc-Savoie, M., et al., Structure-guided evolution of cyan fluorescent proteins towards a quantum yield of 93%. *Nat Commun* 3, 751 (2012). <https://doi.org/10.1038/ncomms1738> and Ai HW, Henderson JN, Remington SJ, Campbell RE. Directed evolution of a monomeric, bright and photostable version of Clavularia cyan fluorescent protein: structural characterization and applications in fluorescence imaging. *Biochem J*. 2006 Dec 15;400(3):531-40. doi: 10.1042/BJ20060874). At higher FRET efficiencies, the mTFP1 lifetime approaches 0.7–0.9 ns, which overlaps cellular autofluorescence lifetimes (approximately 0.4–0.8 ns). The mTurquoise2 lifetime remains further from this background. This does not change the cascade mechanism; it affects measurement robustness. For this reason, where a generic universal donor was required, we used mTurquoise2.

Our thinking was that, for the purposes of this manuscript, which demonstrates a feed-forward structural prediction of intramolecular distances to predict interactions, a state-of-the-art protein system was appropriate to enable others to adopt the technique for their own purposes, as one would design such a sensor from scratch today. For

biological exemplification, we selected a biosensor which has been extensively characterised and optimised for force measurement within vinculin based on an mTFP1-mVenus pairing (Grashoff et al., 2010). Whilst it is possible to sub-clone the biosensor with mTurq2, the subsequent and extensive characterisation which would have been required to re-validate this sensor would be out of all proportion to its impact on the Cascade-FRET exemplification. In fact, we believe that re-engineering the entire tension sensing module would compromise both comparability with the established field and interpretability of the results presented in this manuscript. For this reason, we retained the original sensor and tested the force dependent binding of mScarlet-RIAM to vinculin. This insured that the biological exemplification allowed rigorous cross-validation with the current literature, whilst illustrating the Cascade-FRET principle in a relevant context. In addition, the use of a widely used and validated FRET donor (mTFP1) makes the findings of this paper more relevant to the existing field.

To avoid confusion, we have clarified the figure legend and Methods to state that the VincTS sensor is measured with mTFP1, and we include a brief structural analysis demonstrating donor-agnostic geometry.

With regard to the mVenus G68A mutant, its use was only necessary in the single-chain cascade construct (Figure 1), where a non-fluorescent, non-absorbing mVenus variant was required to control distance and prevent unwanted energy transfer between mTFP1 and mScarlet. In the cellular experiments in Figures 2 and 3, appropriate controls were already included using vinculin-mVenus and vinculin-mTFP1 alone, so incorporating mVenus G68A was unnecessary. Since structurally a dark mVenus construct is not required to maintain separation between mTFP1 and mScarlet (as in the model), incorporation of a mVenusG68A mutant is unnecessary. In this context, RIAM binds Vinculin close to the n-terminus which is significantly prior to the location of the sensor unit within Vinculin.

In regards the localisation of the force within focal adhesions, the reviewer is of course, completely correct. To resolve the distribution of force across the focal adhesion would invoke appropriate super-resolution or single-molecule tracking techniques (see for example, Schlichthaerle, T., Lindner, C. & Jungmann, R. Super-resolved visualization of single DNA-based tension sensors in cell adhesion. *Nat Commun* **12**, 2510 (2021). <https://doi.org/10.1038/s41467-021-22606-1>), however, for the purposes of this manuscript we did not explore these concepts which are significantly beyond the scope of this report. To reassure the reviewer, we have included Supplementary Figure S13, which shows segmented focal adhesions, whole-cell, and excluded focal adhesion masks for the same data presented in Figure 2. This shows the interaction for different sub-cellular compartments and reveals that the interaction is not limited to focal adhesions.

We thank the reviewer again for their identification of an oversight in the materials and methods section of the manuscript regarding the Western blot co-IP shown in Figure 2(b). We have expanded this section of the materials and methods between lines 404 and 422 of the main text.

Regarding the lack of detail regarding the vinculinTL and vinculinTS constructs, Lines 236-239 in the main text explain that the vinculin tensions sensor constructs were made previously (Grashoff, C. et al. Measuring mechanical tension across vinculin

reveals regulation of focal adhesion dynamics. Nature 466, 263-266 (2010)) and have been used characterised by others (Brenner, M. D. et al. Spider Silk Peptide Is a Compact, Linear Nanospring Ideal for Intracellular Tension Sensing. Nano Lett 16, 2096-2102 (2016) and Becker, N. et al. Molecular nanosprings in spider capture-silk threads. Nat Mater 2, 278-283 (2003). <https://doi.org/10.1038/nmat858>).

Once again thank you to the reviewer for identifying inadequate reporting of errors and significance in Fig. 2(b). We have now added the statistical significance to the Figure as requested. Small typographical errors regarding references have been corrected as requested.

We believe the manuscript has been substantially strengthened as a result of these revisions, and we trust the concerns of the reviews have been wholly satisfied and the manuscript is now suitable for publication in *Communications Chemistry*.

We are grateful again to the reviewers and to you for your careful consideration of our work.

Sincerely,

Conor A. Treacy, PhD

Reviewer 1

Comment: Typographical Error on Page 19 - The phrase "Supplementary tables" is unnecessarily repeated on page 19 and should be corrected.

Action taken: We thank the reviewer for spotting this typographical error which we have removed.

Comment: Lifetime Color Palettes in Figure 3.

The fluorescence lifetime color palettes in Figure 3 appear inconsistent. In particular, panels 3b and 3c use different palette scales within the same figure. This may be confusing for readers and should be standardized to improve clarity.

Action taken: LUTs have been changed and are now all the same throughout figures 2 and 3

Comment: Orientation Factor in R_0 Calculations.

The orientation factor (κ^2) plays a critical role in determining FRET efficiency and, by extension, molecular distances. The manuscript does not specify the assumed value. Given the use of flexible six-amino-acid GGSGGS linkers and the evidence of considerable fluorophore mobility shown in Figure S5, a value of $\kappa^2 = 2/3$ (random orientation) seems appropriate. While the FRET-derived distances correlate well with TEM data (Figure S9), it would strengthen the methods section to explicitly state the orientation factor used in R_0 calculations.

Action taken: New sentences added between lines 134 and 136 describing the rotational freedom of attached fluorescent proteins having a kappa-squared value of 2/3

Reviewer 2

Comment: Experiments related to Fig. 2c require an additional negative control using a focal adhesion protein that does not bind to RIAM, such as Paxillin-GFP. It is anticipated that paxillin-GFP will not undergo FRET when co-expressed with RIAM-mScarlet

Action taken: Paxillin-GFP was used in conjunction with mScarlet-RIAM to show no FRET and therefore no interaction between Paxillin and RIAM as requested.

Comment: Likewise, experiments involving the tension sensor (Fig. 3b) would benefit from additional control constructs, such as paxillin-mTFP and paxillin-mTFP-mVenus.

Action taken: We have not carried out this experiment. Given the GFP-Paxillin + mScarlet-RIAM experiment showed no interaction between paxillin and RIAM there is no need to repeat this with a tension sensor as it would not act as a control for this already well-established system.

Comment: Figure 2(d) is difficult to understand. Why is the FRET efficiency similar (not significant) for VincTL and VincTS? Wouldn't you expect to see a significant difference in the blots presented on the left-hand side of the figure? Could the force distribution maps below the VincTL and VincTS FRET distributions be mixed up? I would have expected to see low force (high FRET) in the VincTL sample and higher force (low or heterogeneous FRET) in the VincTS sample. However, this would contradict the non-significant FRET efficiency values on the left

Action taken: We thank the reviewer for their careful inspection of the figure. The labels were indeed misplaced, this has now been rectified and we hope that this clears up any confusion.

Comment: Only lifetime images of focal adhesions are shown, but the schematic model also shows RIAM binding in the cytoplasm. It would be interesting to determine whether vinculin binds to RIAM in the cytoplasm. Therefore, it would be beneficial to present FRET values for areas outside of focal adhesions in a separate figure (see also comment below).

Action taken: Supplementary figure S13 now includes the segmented focal adhesions, cytoplasm and inverted focal adhesion masks for the same data presented in Figure 2. This shows that the interaction is not just focal adhesion-based.

Comment: The model does not reflect current knowledge about RIAM. According to this knowledge, Rap1 first recruits talin at the membrane, forming a ternary complex with RIAM. Furthermore, the model presented here does not consider that the RIAM TBS site and the talin-vinculin binding site (VBS) bind to the same region of vinculin. This precludes the possibility of RIAM binding to both talin and vinculin simultaneously, as shown in the model figure. Instead, the model suggests that the main RIAM-vinculin interaction site may be in

the cytoplasm, which could prevent vinculin from binding to talin. If vinculin can be recruited to focal adhesions by other proteins without binding to talin, one might imagine direct vinculin-RIAM binding also in focal adhesions

Action taken: The section of the discussion relating to Figure 4 (lines 320-333) has been rewritten to explicitly explain that our results do not contradict previous work, which showed that RIAM and vinculin binding to talin are mutually exclusive. Furthermore, we have modified the diagram in Figure 4 to better illustrate this and avoid any potential confusion.

Comment: Fig 2 figure legend had the title duplicated.

Action taken: Duplication has been removed

Comment: Several of the reference numbers appear to be out of line with the in-text citations.

Action taken: We again thank the reviewer for their careful reading; the issue has been resolved

Comment: The title does not make sense if RIAM binds vinculin in a force-independent manner.

Action taken: After some consideration we agree with the reviewer and have updated the title to reflect the force independent manner of RIAM-Vinculin binding

Reviewer 3

Comment: It is inappropriate that the authors used mTurq2 in Fig. 1, while mTFP1 was used for FRET measurements in Figs. 2 and 3. The same fluorescent protein, mTurq2, should be used to demonstrate the development of the three-color FRET method and its application. In addition, the mVenusG68A mutant used in Fig. 1 was not used in the cellular experiments in Figs. 2 and 3.

Action taken: We agree that consistency is desirable. The VincTS biosensor, designed using mTFP1 and published in Nature Methods in 2010 (Grashoff et al., 2010), has been thoroughly validated and is widely used. We therefore maintain mTFP1 as the donor in all

VincTS measurements, which preserves continuity with the established literature and enables us to utilise the validated stiffness measurements for cellular force calculations. We therefore retain mTFP1 in VincTS experiments for continuity with the established sensor, and we clarify in the text that Supplementary Figure S7 is a schematic representation. We add a brief structural analysis showing that the cascade works equivalently with either donor see lines 175-180 for results of this and lines 448-646. With regard to the mVenusG68A mutant, its use was only necessary in the context of the single-chain cascade construct (Figure 1), where a non-fluorescent, non-absorbing mVenus variant was required to control for distance and prevent unwanted energy transfer between mTFP1 and mScarlet. In contrast, for the cellular experiments in Figures 2 and 3, appropriate controls were already included using vinculin–mVenus and vinculin–mTFP1 alone, making the incorporation of a mVenusG68A mutant unnecessary.

Comment: In Fig. 2c and d, Fig. 3b and c, measuring FRET efficiency and force over the entire cell yields poor spatial resolution. Within a single cell, force exerted on focal adhesions is heterogeneous, and even within individual focal adhesions, force distribution can vary. Averaging across the whole cell obscures these local variations in force and FRET signals. The analysis should instead be restricted to subcellular regions, particularly focal adhesions.

Action taken: Supplementary figure S13 now includes the segmented focal adhesions, cytoplasm and inverted focal adhesion masks for the same data presented in Figure 2. This shows the interaction for different sub-cellular compartments and reveals that the interaction is not limited to focal adhesions. The analysis of force within focal adhesions would require super-resolution and is beyond the scope of this work.

Comment: The Materials and Methods section does not provide sufficient information. The description of a western blot of the co-IP of EGFP-vinculin with RIAM and Talin (Fig. 2b) is lacking.

Action taken: We thank the reviewer for this suggestion and have expanded this section of the materials and methods between lines 397 and 415 of the main text.

Comment: No detail of the vinculinTL and vinculinTS constructs were given.

Action taken: Lines 232-234 in the main text explain that the vinculin tensions sensor constructs were made previously (Grashoff, C. et al. Measuring mechanical tension across

vinculin reveals regulation of focal adhesion dynamics. Nature 466, 263-266 (2010)) and have been used characterised by others (Brenner, M. D. et al. Spider Silk Peptide Is a Compact, Linear Nanospring Ideal for Intracellular Tension Sensing. Nano Lett 16, 2096-2102 (2016) and Becker, N. et al. Molecular nanosprings in spider capture-silk threads. Nat Mater 2, 278-283 (2003). <https://doi.org/10.1038/nmat858>).

Comment: The reference numbers given in the main text are incorrect (I noticed this around Ref. 38, p2).

Action taken: We thank the reviewer for their careful reading; this issue has been resolved

Comment: Fig. 2b, right panel, statistical significance should be shown.

Action taken: Fig. 2b now includes the missing statistical significance